# The Design and Verification of an Active SAMSR Ultrasonic Guided Wave Monitoring System with Ultra-Low Crosstalk

**DOI:** 10.3390/s20030898

**Published:** 2020-02-07

**Authors:** Wendong Xue, Mengjiao Yang, Deming Hong, Di Wu, Yishou Wang, Xinlin Qing

**Affiliations:** 1School of Aerospace Engineering, Xiamen University, Xiamen 361005, China; yangmengjiao@stu.xmu.edu.cn (M.Y.); hongdeming@stu.xmu.edu.cn (D.H.); wangys@xmu.edu.cn (Y.W.); xinlinqing@xmu.edu.cn (X.Q.); 2China Academy of Launch Vehicle Technology, Beijing 100076, China; wudibuaa@163.com

**Keywords:** ultrasonic guided wave, single-actuation and multiple-simultaneous-reception (SAMSR), ultra-low crosstalk, matrix switcher

## Abstract

Due to long propagation distance and high sensitivity to a variety of damages, ultrasonic guided wave technologies have been widely applied in the damage detection or health monitoring of pipe networks and large plate-like structures. However, there are two important problems to be solved when applying this technology; namely, the large scanning time required for monitoring large-scaled structures and the serious crosstalk between the actuation and receiving signals, especially when monitoring hot-spot regions. Therefore, this study mainly designed key parts, such as the matrix switcher and attenuation circuit. The single-actuation and multiple-simultaneous-reception (SAMSR) mechanism based on an analog switching matrix and a low noise charge amplifier circuit was designed and integrated with the SPI control bus to shorten the scanning time. Moreover, a two-stage attenuation circuit with an interlocking isolation structure is presented to effectively isolate the receiving signals from the actuation signals to obtain ultra-low crosstalk even under a high voltage actuation source. In this study, the designed matrix switcher and other components were integrated into the developed ultrasonic guided wave monitoring system. Several experiments were conducted on a stiffened composite structure to illustrate the effectivity of the developed SAMSR ultrasonic guided wave system by comparing the signals collected with those from a commercial ultrasonic guided wave system.

## 1. Introduction

As an efficient and flexible online/offline damage inspection and diagnosis method, ultrasonic guided wave technology has already been widely applied in the damage inspection of engineering structures such as pipes, railways, plate-like structures and connected structures [1,2,3]. Generally, piezoelectric transducers (PZTs) which are light-weight, small in volume and have a wide frequency response are utilized to actuate and receive ultrasonic guided waves. PZTs can be not only bonded to the surface of different structures, but also embedded into composite materials. Therefore, the ultrasonic guided wave technology based on PZT sensor networks has been showing potential for an application in the structural health monitoring (SHM) of large aircraft structures and other hot-spots [4] where there are structural parts or regions with high stress, such as bolted and adhesive joints.

The hardware system of ultrasonic guided waves is the connecting and processing center of piezoelectric sensor networks and damage diagnosis algorithms. The system’s core function is to excite the piezoelectric sensor to generate ultrasonic guided waves in the structure and collect and process the guided wave signal perceived by the other sensors. The hardware system of ultrasonic guided waves includes, mainly, a signal generator, a signal collector, a sensor network and a matrix switcher. The switcher is used to connect these three components. In recent years, many research institutes and companies have developed some ultrasonic guided wave monitoring systems, and conducted preliminary application verification. For example, the US company ACELLENT has developed the commercial ultrasonic guided wave detector ScanGenie III, which adopts pitch- catch data scanning mode (single-actuation and single-reception, SASR) and can support the input of up to 128 channels in the sensor array. Yuan Shenfang et al. [5] designed and developed an integrated health monitoring and scanning system of aircraft structures based on ultrasonic guided waves, in which a pitch-catch working mode and the input of an up to 24-channel sensor array was realized and supported. Fu et al. [6] designed and developed an energy-saving distributed acquisition system based on wireless sensor network. Each node of the system supports an 8-channel sensor array input. In the system, an analog multiplexer and an amplifier are used to attenuate the crosstalk signal generated by the actuation, and good results are obtained under the actuation signal of no more than 6V. Zheng et al. [7] designed an EMAT (electromagnetic acoustic transducer) system to circumferentially and uniformly excite a pure mode Lamb wave which is used to detect the hole defects on the surface under test. However, due to the existence of crosstalk, the sense’s first arrival signal may add to the crosstalk signal, and even be lost when ADC saturation happening, so there is a blind area for damage detection, which cannot be eliminated. To sum up, at present, to achieve reliable application of ultrasonic guided waves for monitoring large-scale structures, such as aircraft or hot-spot areas, there are still some challenging problems to be solved, such as signal crosstalk between actuation and acquisition in sensor networks under high voltage actuation, and synchronous acquisition to realize the fast scanning mode of SAMSR [8]. 

Crosstalk mainly occurs in active ultrasonic guided wave detection systems, especially in systems where actuation and acquisition sensor networks are shared. The higher the actuation signal voltage, the more obvious the crosstalk phenomenon is. In the application of some high viscoelastic materials, such as composite materials, it is often a requirement to generate ultrasonic guided wave signals with high signal to noise ratios (SNRs) brought by sufficient vibration of PZT sensors with high actuation voltage. Generally, actuation voltage with maximum amplitude within ±70 V is often selected [9,10,11]. For example, the optimal voltage amplitude of actuation signal of commercial use detector ScanGenie III is between ±50 V and ±70 V. Therefore, the crosstalk problem is particularly serious. When the distance between actuation sensor and acquisition sensor is very close, especially in the inspection of hot-spot regions, the crosstalk signals and the effective signal received often add together due to the complexity of detection areas, and there are even saturated signals, which lead to signal loss, seriously affecting the inspection accuracy [7,12,13,14]. Although the crosstalk also exists in phased array ultrasonic detection systems, it would not affect the system’s detection accuracy, which can be ignored directly, because all of the elements start to collect echo signals after the end of the actuation signal in the phased array system. The phased array ultrasonic detection systems mainly focus on the delay time compensation, which is different from how the ultrasonic guided wave detection systems work [15,16,17].

In addition, as the monitoring area increases, the number of sensor networks will increase rapidly, as do the times of actuating and acquitting in a damage scan cycle. However, in the scanning process, to ensure that the signals between the primary and secondary actuation do not affect each other, each actuation must wait for the complete disappearance of echo ultrasonic guided waves generated by the former actuation. Therefore, when collecting a large number of sensor channels in the traditional pitch-catch data acquisition mode, the scanning cycle is long, and thus the overall monitoring efficiency may be affected by environmental changes [18]. In fact, when any sensor is treated as an actuation source, the other sensors on the scanning path also can receive ultrasonic guided wave signals, and thus, when making an arbitrary sensor the actuation source, make the other multichannel sensors a synchronization acquisition source. In this way, a SAMSR operation mode is realized, which can effectively improve the efficiency of detection and reduce the effects of environmental change.

Aiming at the application demand of ultrasonic guided wave technology based on piezoelectric sensor networks in structural health monitoring, this paper designs an active SAMSR ultrasonic guided wave monitoring system with ultra-low crosstalk, and focuses on the design of the matrix switcher and other core components. These key components mainly include a 64-channel and single-actuation and multiple-simultaneous-reception analog switch matrix based on SPI (serial peripheral interface) control bus and the four-channel low noise charge amplifier circuit, and a two-stage attenuation circuit and a six-layer shielding circuit board based on the interlock isolation structure, where the actuation is shielded by ground envelope and highly isolated from the received signal. Moreover, the designed matrix switcher and other core components are integrated into the developed ultrasonic guided wave system; the signals collected by the system and those of commercial use ultrasonic guided wave system are compared and analyzed; and the damage location experiment of hot-spot areas is also carried out in this paper. The major advantages of this system are listed as follows:(1)Ultrasonic guided wave signals with ultra-low crosstalk can be obtained even under high voltage actuation;(2)The damage detection efficiency is increased effectively by a SAMSR operation mode;(3)The effect of crosstalk is minimized as much as possible to realize the precise damage location of a hot-spot;(4)The large-scaled structures can be monitored by the developed system due to its strong scalability. The attenuators at all levels are controlled by the same control signal from the attenuation design of interlocking isolation structures. When the attenuation series increase, the switching time would not increase correspondingly. Theoretically, the analog switch matrix based on SPI control bus structures can realize the expansion of any multichannel sensor network and the synchronous acquisition of any multichannel.

## 2. Design of the Ultrasonic Guided Wave Detecting System Based on PZTs Network

### 2.1. Basic Compositions and Principles of the Ultrasonic Guided Wave Active Detecting System

An ultrasonic guided wave active monitoring system is commonly composed of a sensor network; signal actuation and acquisition hardware; and signal control and processing software [4]. The electronic hardware system mainly includes a signal generator, signal collector, power amplifier, charge amplifier and a matrix switcher for controlling the switching of actuation and acquisition sensors, as shown in Figure 1. The signal generator generates a 5-cycle Hanning windowed toneburst. The actuation signal is input into the matrix switcher when the power amplifier is amplified to the power suitable for the actuation of PZT. The matrix switcher switches the corresponding actuation sensor and the acquisition sensor, and sends the acquisition signal back to the charge amplifier, which is then amplified and collected by the signal collector. In said way, a cycle of signal actuation and acquisition is completed. 

Ultrasonic guided wave active monitoring technology uses a PZT sensor mounted on the surface of the structure being tested to actuate ultrasonic guided waves, and then the wave’s signal is transmitted within the structure and finally received by other sensors. If the ultrasonic guided wave interacts with the damage during the propagation, the received ultrasonic guided wave then contains damage information which can be obtained by comparing it with the ultrasonic guided wave when the structure is not damaged [19]. Therefore, a guaranteed high SNR and high repeatability of the signal is the premise and key of the subsequent signal analysis and damage location; i.e., ultrasonic guided wave monitoring has higher requirements when designing the hardware system. 

### 2.2. Analysis of Problems in the Ultrasonic Guided Wave System

Focused on the crosstalk encountered and the SAMSR operation mode, this paper analyzes mainly, the reasons behind it and its design basis.

#### 2.2.1. Analysis of Crosstalk

In the ultrasonic guided wave monitoring system, in order to reduce the number of sensors, the actuation and acquisition sensors basically share the same set and they are switched by a matrix switcher. Effective isolation can be achieved by a semiconductor switching device at low voltage, and crosstalk signals can be suppressed [6]. However, semiconductor devices have parasitic capacitance, and large leakage current can be produced by their substrates at higher voltage. However, the frequency of actuated wave is generally around 50–1000 kHz which is in a relatively high frequency range. Therefore, it is easy for it to pass through the isolation by the parasitic capacitance and produce induction signals in the collection channel, which is one of the sources of crosstalk signals. The strength of the induction signal increases with the increase of the actuation voltage, which is superimposed with acquired ultrasonic guided wave signals and amplified after passing through the charge amplifier, resulting in the distortion and even saturation of the ultrasonic guided wave signals. As displayed in Figure 2a,b, the actuation signals of sensors and (c) and (d) are the acquisition signals after being amplified by the charge amplifier. From Figure 2c, it can be clearly seen that at the beginning of the acquisition signal, the crosstalk signal with the same waveform of actuation signals appears, and the signal amplitude is very high. When the actuation and the acquisition sensors are very close to each other, especially when applied to the monitoring of hot-spot areas, ultrasonic guided wave signals would be detected by the acquisition sensor before the actuation period finishes, as shown in Figure 2d. The ultrasonic guided wave signal detected by the acquisition sensor is superimposed on the crosstalk signal on the acquisition channel, which distorts the signal and makes it impossible to obtain the true and effective signal, thereby affecting the accuracy of damage location. 

When the actuation and acquisition system are completely separated in the design of hardware system, there is no current loop between the actuation and acquisition signal, so the crosstalk problem can indeed be solved. However, because sensors cannot be shared, the number of sensors would then be multiplied, which increases the complexity of the system. If sensors are to be shared, the key of the system lies in the design of matrix switcher; i.e., how to achieve effective isolation between actuation and reception.

#### 2.2.2. The Principles of SAMSR Operation Mode

The traditional SASR operation mode is displayed in Figure 3. Taking monitoring areas which consist of five PZTs, in the whole scan path, when PZT 1 is selected as the actuation source and the other four are reception sources, the data acquisition steps are as follows: The primary actuation of PZT 1 is followed by the acquisition of PZT 2. Wait until the echo ultrasonic guided wave disappears. PZT 1 is actuated for the second time and followed by the acquisition of PZT 3. Wait until the echo disappears. This cycle continues until all four reception sources have collected all the data. Therefore, PZT 1, when used as the actuation source, needs acquisition and delay four times to complete damage scanning. To complete the scanning of the whole detection area, five PZTs are needed due to the actuation of the source one by one, so a total of 20 rounds of actuation and delay are required. As the number of sensors increases, the number of actuation and delay times also increases exponentially, so the damage scanning cycle is long and the system monitoring efficiency is low. Moreover, due to the long duration of sampling, the signal would be substantially affected by environmental changes, affecting the accuracy of damage location.

As can be seen from the SASR signal sampling process, when PZT 1 is first actuated as the actuation source, all the other PZTs will receive ultrasonic guided wave signals simultaneously. If signals from multiple sensors can be collected into the system at the same time, the number of actuation and delay times can be greatly reduced, thereby reducing scanning time. Therefore, this paper designs a SAMSR operation mode; i.e., when the same five PZTs are used as the actuation source separately, the whole damage scan cycle requires only five rounds of actuation and delay. The core problem of this design also lies in the design of matrix switcher to make it possible that any sensor can be switched to actuation source, and the other four sensors can be switched to acquisition channels, and the four channels can sample synchronously.

## 3. Hardware Design

### 3.1. Overall Design

The overall design of the active SAMSR ultrasonic guided wave with ultra-low crosstalk system is displayed in Figure 4. The system includes mainly a main controller, a power amplifier, a matrix switcher and PZTs sensor networks. Among these, the main controller includes a HMI (human–machine interface), a signal generator and four-channel oscilloscope acquisition modules. The signal generator outputs a 5-cycle Hanning windowed toneburst signal while triggering the oscilloscope acquisition module through TRIG signal to sample synchronously. The toneburst signal passes through the power amplifier and generates actuation signals to output to the matrix switcher. The matrix switcher connects the signal actuation channel with the designated PZT and connects the four other designated PZTs to the signal acquisition channel under the scanning demand from the main controller, and outputs ultrasonic guided wave signals to the acquisition module of the four-channel oscilloscope of the main controller after being amplified by the independent charge amplifiers.

Each sensor channel of the matrix switcher is designed with a two-stage attenuator to highly isolate the actuation from the reception channels. The primary attenuator is composed of a high speed SPDT (single pole double throw) reed relay. An SPDT reed relay has a set of mechanical inter-block with NO (normally open) contacts and NC (normally closed) contacts. The switching time of it is generally less than 2 ms, and the reed relay contacts are protected by inert gas, preventing them from oxidizing and giving them long contact life. As shown in Figure 4, the SPDT reed relay is not driven by default, and the NC contact is closed. The PZT is connected with the reception channel by default and is completely separated from the actuation channel, forming a highly isolated state. The STM32 controller on the matrix outputs the address of the sensor to be selected as the actuation source through the address bus. After decoding by a 6-64 decoder, the corresponding SPDT reed relay is driven to make the NC contact open, and the NO contact closed. The corresponding PZT sensor automatically switches from the reception channel to the actuation channel under interlock control, and disconnects physically from the reception channels to form high isolation, which, at the same time, also prevents the high voltage actuation signal from being connected to the reception channel and avoids the breakdown of low voltage devices. After passing through the primary attenuator, the actuation and reception channels are almost completely isolated. However, the reed relay still has a small capacitance effect due to the small spacing between the reed and the contacts, which makes it possible for high voltage and high frequency actuation signals to form induction on the NC contact of the reception channel and generate weak crosstalk. Therefore, a secondary attenuator is designed to be on the actuation channel. The secondary attenuator is composed of a high-speed SPST reed relay with NO contacts. The drive powers of the two attenuators reed relays are controlled in parallel by the same control signal, so the switching time will not increase. Therefore, the addition of the secondary attenuator actually increases the distance between the actuation and the reception contact without increasing the switching time, which reduces the effect of parasitic capacitance further so that the crosstalk signal decays exponentially. When the actuation signal has higher voltage or frequency requirements, more stage attenuators can be designed to suppress crosstalk signals.

A four-channel 64 × 1 multiplexers is designed for the ultrasonic guided wave reception channels of the matrix switcher, forming an analog switching matrix. In the design of the primary attenuator, the sensors are connected to reception channel by default; therefore, except for the actuation sensor, all the other sensors are connected with the 64 × 1 multiplexer of each reception channel by default. The STM32 controller on the matrix switcher connects all channel multiplexers in series through the SPI control bus, forming a daisy-chain control structure, which allows any four sensors to be selected as the reception sensor and makes the extension of more simultaneous reception channels more convenient. Because the ultrasonic guided wave signal acquired by the sensor is a very weak charge signal and the output impedance is high, a charge amplifier is designed on each reception channel.

### 3.2. Realization of the Matrix Switcher Module With Ultra-Low Crosstalk

This paper takes the sensor input of 64-channel as an example to achieve the SAMSR operation mode with the frequency of the actuation signal ranging from 50 to 700 kHz and an voltage amplitude in the range of ±70 V. Feasibility verification was carried out and the circuit schematic diagram is shown in Figure 5. The primary attenuator adopts an SPDT reed relay of the DIP05-1C90 series from MEDER. The maximum switching time of it is less than 1.5 ms and the isolation resistance is up to 10^9^ GOhm. The secondary attenuator adopts an SPST reed relay of the CRR05 series; the maximum switching time of it is less than 0.6 ms; and the isolation resistance is up to 10^14^ Ohm. As the control signals of the primary and secondary attenuator are connected in parallel, the overall switching time is less than 1.5ms. In the scanning process, the delay of milliseconds between the two excitation signals is also required to wait for the echo to disappear, so the reed relays switching speed would not affect the damage scanning cycle. The drive signal of the reed relay is provided by two relay driver chips ADG732 (32 to 1) directly, which constitutes a multiple selector (64 to 1). STM32 controller outputs a 6-bit address bus, controlling the decoder and realizing 6–64 address decoding. The analog switch matrixes of the collection channels are composed of 32 pieces of ADG738 analog multiplexers with low resistance in series, which is controlled by the SPI bus of STM32 to achieve four-channel 64–4 multiplexers. By controlling the four-channel multiplexers and taking any four sensors except the actuation as the collection sources, the four channels’ ultrasonic guided wave signals could be received synchronously and output to the independent low noise charge amplifier circuit.

Figure 6 shows the comparative analysis of crosstalk signals after the primary attenuator and secondary attenuator. Figure 6a shows the actuation signal with an amplitude about ±70 V; Figure 6b shows the ultrasonic guided wave signal acquired by a commercial instrument isolated by a high voltage CMOS switcher and the amplitude of the crosstalk is big; Figure 6c shows the signal acquired after being isolated by the primary attenuator, and the crosstalk of it is already rather small (the amplitude of it is less than 4% of that of effective signal); Figure 6d shows the signal acquired after being isolated by the secondary attenuator, and the crosstalk of it almost disappears and can be easily filtered such as Savitzky-Golay filter algorithm

### 3.3. Design of a Low Noise Power Amplifier

Since the principle of damage detection and location for an ultrasonic guided wave system is to compare the changes of signals before and after the damage, the noise and phase shifts of acquired signals will have a great impact on the damage detected accuracy. The received signal of the PZT is a weak charge signal, which needs to be amplified by the charge amplifier before it can be accurately collected by the data acquisition system. In this paper, according to the performance requirement of 1M bandwidth acquisition signal, OPA132 chips are adopted to design the primary power amplifier circuit and the primary voltage amplifier circuit (as shown in Figure 7), and the amplification is designed to adjustable by DIP switchers. 

In order to prevent the acquisition signal from generating different phase shifts at different frequencies, which would affect the repeatability and accuracy of damage detected accuracy, no hardware filtering is set in the signal collection channel; the phase-shift-free Savitzky-Golay algorithm is used in the software for filtering instead. In terms of hardware, the Π filter and the LDO (low dropout regulator) filter are used, which puts the V_p-p_ of the power supply noise under 4.4 mV. As displayed in the diagram of Figure 8a, the CH1 shows the output wave after being filtered by the Π filter and CH2 shows the output power ripple after being filtered by LDO. When the input of the charge amplifier is connected to the GND, the signal noise (V_p-p_) is suppressed under 6mv, as shown in the diagram of Figure 8b.

To reduce the crosstalk between the actuation signal and the reception signal in the PCB routing, the matrix switch PCB adopts a six-layer structure in which devices are mounted on the top and bottom layer, and the high voltage actuation signal is mainly routed in the bottom layer, and it is surrounded by ground envelope and shielded by a layer of GND. The 64-channel PZT signals are routed in layers 1 to 4, and each adjacent channel is distributed in different layers. The layered design ensures the shielding and isolation between different signals, and each signal is shielded by GND, which ensures that each signal is transmitted independently and is not disturbed by electromagnetic interference. As shown in Figure 9, there are six modules in the PCB board, including a power module (1), a MCU module (2), a relays’ address selector and drive module (3), a two-stage attenuation relay module (4), a 64 to 4 analog switcher module (5) and a charge amplifier module (6).

## 4. Experimental Verification

Based on the PXI platform of NI, this paper designs and builds an ultrasonic guided wave detecting system into which a SAMSR ultra-low crosstalk matrix switcher is integrated. Simulated damage location and identification experiments were carried out on the hot-spot areas of composite materials. The experimental platform is displayed in Figure 10. The main controller of this system is NI’s PXIe-1071 chassis, which is equipped with a CPU card (PXIe-8820), a signal generator card (PXIe-5442) and a multichannel synchronous acquisition oscilloscope card (PXIe-5105). Actuation signals output by the signal generator card after being amplified by the power amplifier have a voltage amplitude within ±70 V, and then they are output into the matrix switcher developed in this paper. The ultrasonic guided wave signal output by the matrix switcher is then amplified and input into NI’s multichannel oscilloscope card. The signal generator card and oscilloscope card are triggered synchronously through the system bus of PXI to ensure actuation output, and also the simultaneous sampling and acquisition of signals.

The composite material board is 450 × 450 × 3 mm in size, with a T-shaped stiffener (height 40 mm) and a thickness of 1.7 mm, which is glued to the main board to simulate the glued connection structure of hot-spot areas [13,20]. The composite material plate is made of 15 plies of T300 woven prepreg with a process of vacuum bag molding. The positions and signal propagation paths of the test board and sensors are shown in Figure 11. The coordinates of the composite material board at the bottom left are (0, 0) and at the top right corner are (450, 450). Eight PZTs (type PZT-5A) are symmetrically arranged on both sides of the connection structure.

### 4.1. Crosstalk Test of the System

Under the same experimental conditions, the ScanGenie III ultrasonic guided wave detector produced by ACELLENT and the proposed system in this paper are used for the crosstalk test of the system respectively. The frequency of the actuation signal center is set at 250 kHz, and different actuation and reception sensors are selected. The acquired signals are shown in Figure 12. It can be seen from the figure that the crosstalk signal of ScanGenie III is relatively big, and when the actuation and the reception sensors are close, as shown in Figure 12b, the crosstalk signals and effective signals are superimposed together; the crosstalk signals of the system developed in this paper are ultra-low and the echo signals can be fully displayed.

### 4.2. System Repeatability Test 

Since the damage location of the ultrasonic guided wave system is based on the comparative analysis of the signals before and after the damage, the repeatability of the signals collected by the system is very important. Displayed in Figure 13 is the same path signals acquired by the proposed system. Figure 13a is the waveform of signals after being actuated twice under the same center frequency 250 kHz, and diagram Figure 13b is the waveform of ΔE, the signal difference of the two actuation signals. It can be seen that the effective signal amplitude is about ±1.9 V; the maximum repeatability error is around ±0.039 V; and the amplitude ratio is around 2.05%, which shows that the system has good repeatability.

Error is calculated by standard mean square error. The system developed in this paper and ScanGenie III are tested with the actuation center frequency as 50–700 kHz. The results are shown in Figure 14. It can be seen that the repeatability of the system designed in this paper is relatively stable within 700 kHz, while the error of ScanGenie III fluctuates greatly.

### 4.3. Simulated Damage Location of Hot Spot Region

Firstly, the experimental plate is scanned at a frequency of 50–700 kHz, and the frequency that maximizes the effective signal amplitude is obtained; i.e., the resonance frequency of PZT and composite material plate. At this frequency, the acquisition signal has the highest SNR. After frequency scan, the resonance frequency in this experiment obtained is about 300 kHz. Therefore, five peak modulated sine waves with a central frequency of 300 kHz are taken as the actuation signal to collect the baseline signal and damage signal respectively. The baseline signal is the signal acquired from all paths under the condition of no damage, and the damage signal is the signal of all the measured paths after the simulated damage when an absorbing adhesive (size Φ25 mm) is placed on the glued connection structure of the surface under testing with the same actuation of the reference signal. Figure 15 shows the signal of path PZT 3-5/6/7/8 when the simulated damage is pasted around PZT 3. After comparison, it can be seen that there is an amplitude or phase difference between the reference signal and the damage signal.

The elliptic weighting algorithm [21,22] is used to analyze the damage location of the acquired signal, the diagnostic image is obtained and the location of the damage point is output, as shown in Figure 16. The brightest part of the color of the diagnosed damage position coincides with the actual damage position; therefore, the location is accurate.

## 5. Conclusions

This paper designed and developed an active SAMSR ultrasonic guided wave monitoring system with ultra-low crosstalk. In this system, the core part, i.e., the 64 × 4 ultra-low crosstalk matrix switcher, was developed and integrated with 4-channel charge amplifiers with low noise and phase shift. The SAMSR operation mode and 64 channel of PZTs input are supported in the developed system. A simulated damage location experiment was carried out on composite materials with hot-spot regions; good results were obtained. The matrix switcher is designed to verify the effect of the design scheme proposed in this paper. By optimizing the selection of chip devices further, higher integration can be achieved. The main contributions of the system include: (1)A two-stage attenuating circuit with an interlocking isolation structure is designed to achieve high isolation between actuation and acquisition signals. Under high voltage (±70 V) actuation, ultrasonic guided wave signals with ultra-low crosstalk can also be obtained.(2)This design supports the input of an up to 64-channel sensor array and adopts the mode of single-actuation and four-simultaneous-reception. In this system, any sensor can be used as the actuation source and the other four as the reception source, which effectively improves the acquisition efficiency.(3)The system has been verified on composite materials of hot-spot areas, for which the effect of crosstalk is almost eliminated and the accurate damage location is achieved.(4)This design has strong expansibility and can be applied to higher actuation voltage and large-scaled area monitoring.

## Figures and Tables

**Figure 1 sensors-20-00898-f001:**
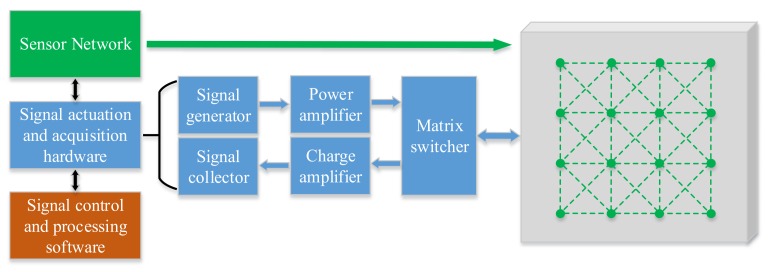
Components of an ultrasonic guided wave detection system.

**Figure 2 sensors-20-00898-f002:**
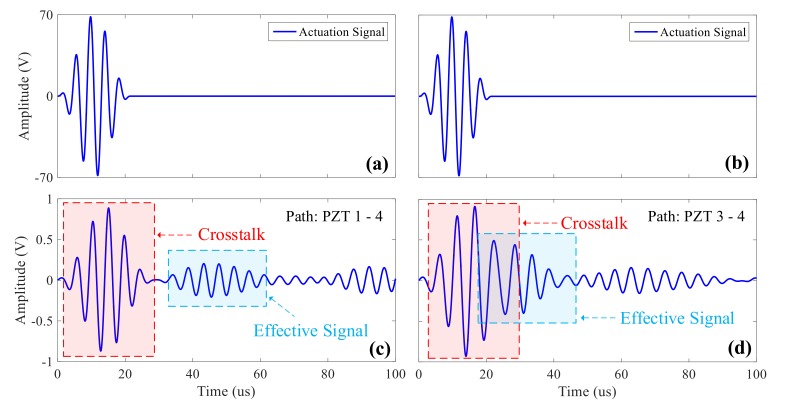
Acquisition diagram of a conventional ultrasonic guided wave system: (**a**,**b**) 190khz actuation signal; (**c**) effective isolation between crosstalk and effective signal; (**d**) overlap between crosstalk and effective signal.

**Figure 3 sensors-20-00898-f003:**
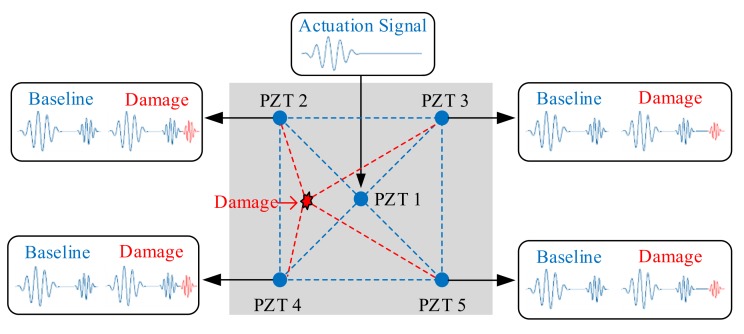
Schematic diagram of damage monitoring.

**Figure 4 sensors-20-00898-f004:**
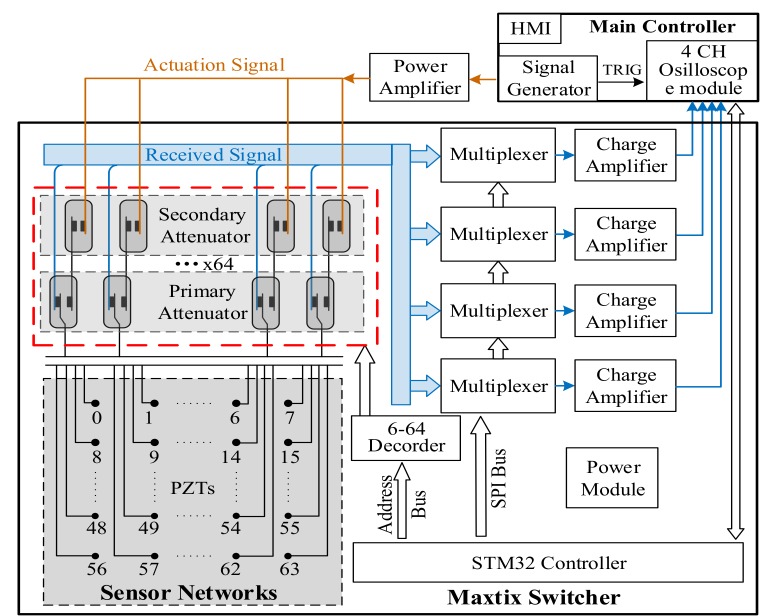
Design diagram of the hardware in the system.

**Figure 5 sensors-20-00898-f005:**
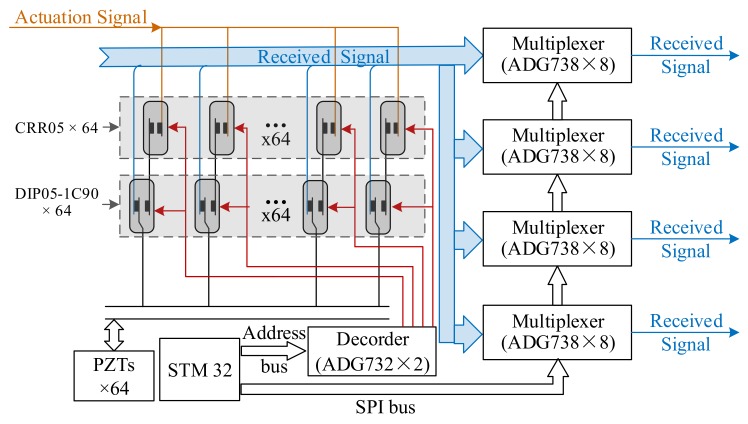
Diagram of matrix switcher module.

**Figure 6 sensors-20-00898-f006:**
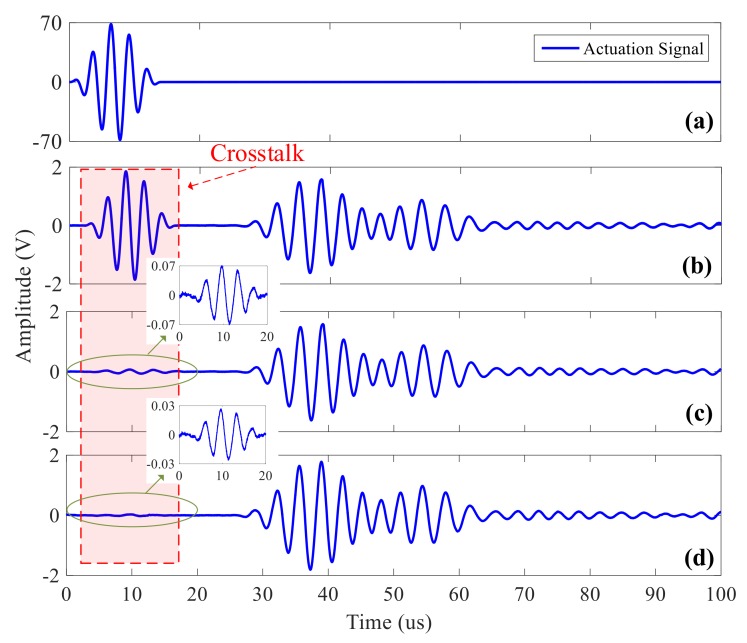
Comparison diagram with the primary attenuator and the secondary attenuator: (**a**) 250khz actuation signal; (**b**) acquisition signal from ex-commercial instrument; (**c**) acquisition signal after being isolated by the primary attenuator; (**d**) acquisition signal after being isolated by the secondary attenuator.

**Figure 7 sensors-20-00898-f007:**
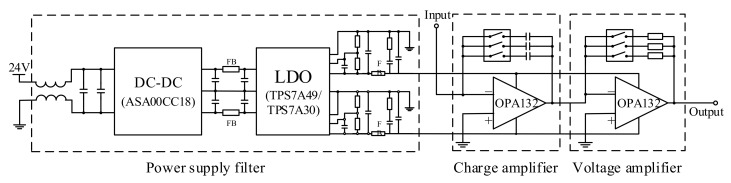
Schematic diagram of the charge amplifier.

**Figure 8 sensors-20-00898-f008:**
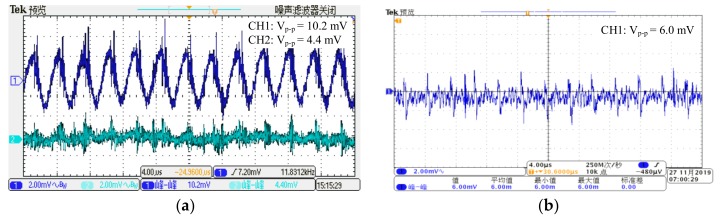
The noise diagram of the charge amplifier: (**a**) the comparison diagram of the power module after filtered; (**b**) the noise of the charger amplifier.

**Figure 9 sensors-20-00898-f009:**
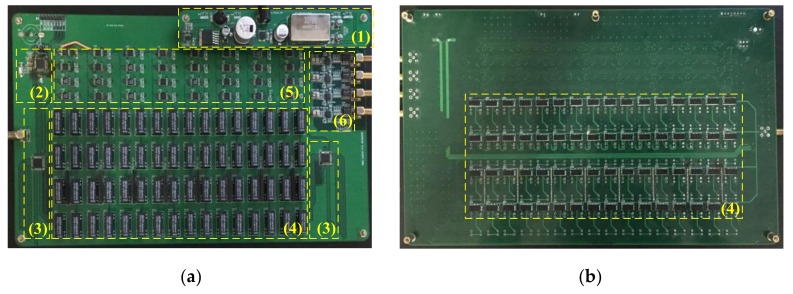
The matrix switcher PCB. (**a**)Top view; (**b**) Bottom view.

**Figure 10 sensors-20-00898-f010:**
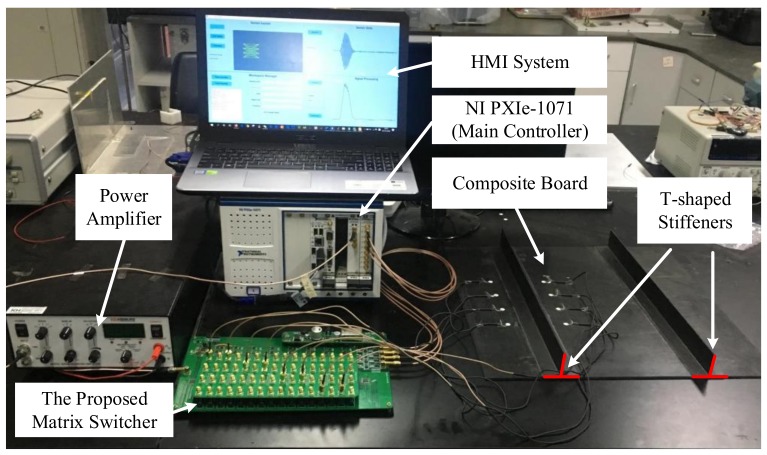
Experiment platform.

**Figure 11 sensors-20-00898-f011:**
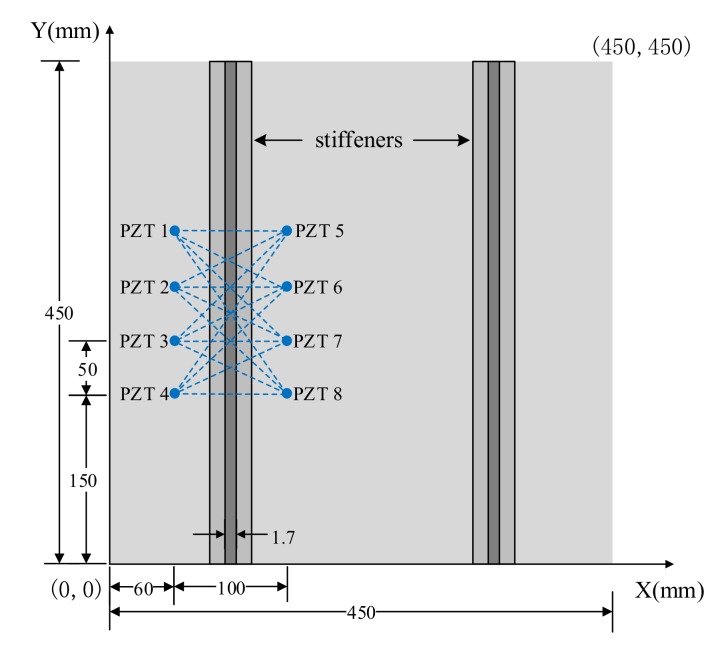
The positions and paths of the sensors.

**Figure 12 sensors-20-00898-f012:**
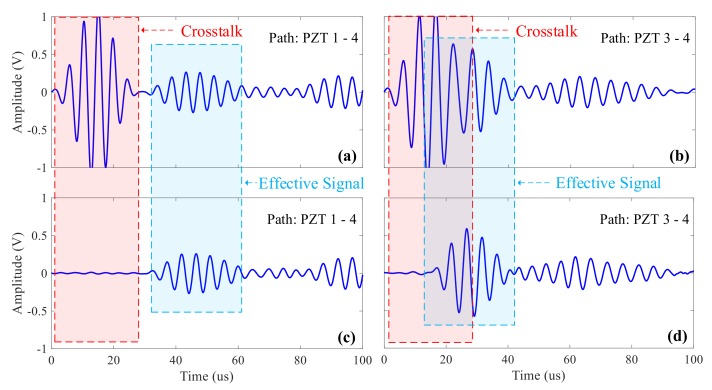
Comparison test of the crosstalk of different systems: (**a**)/(**b**) the waveform acquired by ScanGenie; (**c**)/(**d**) the waveform acquired by the proposed system in this paper.

**Figure 13 sensors-20-00898-f013:**
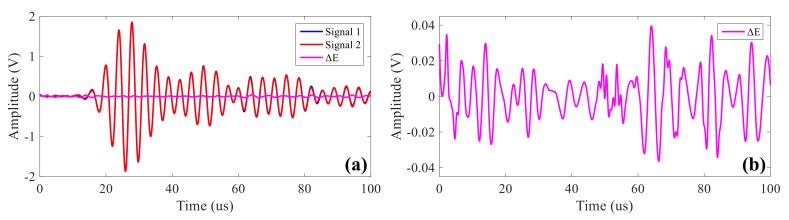
System repeatability test at the frequency of 250 khz actuation signals: (**a**) comparison after two signal acquisitions; (**b**) the difference of the two acquired signals.

**Figure 14 sensors-20-00898-f014:**
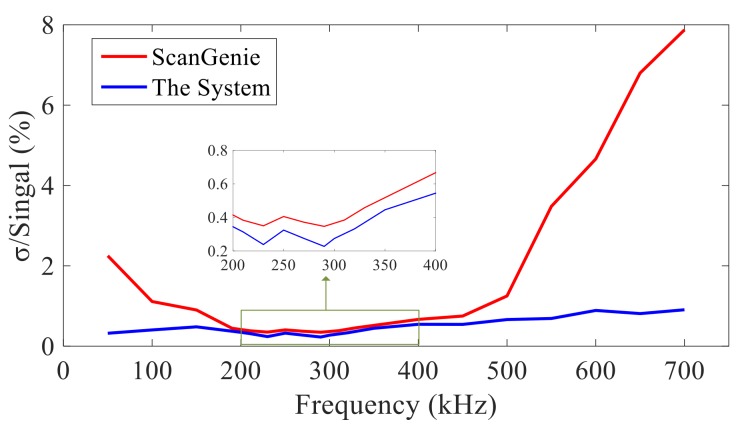
System repeatability test at the frequency of 50–700 kHz actuation signals.

**Figure 15 sensors-20-00898-f015:**
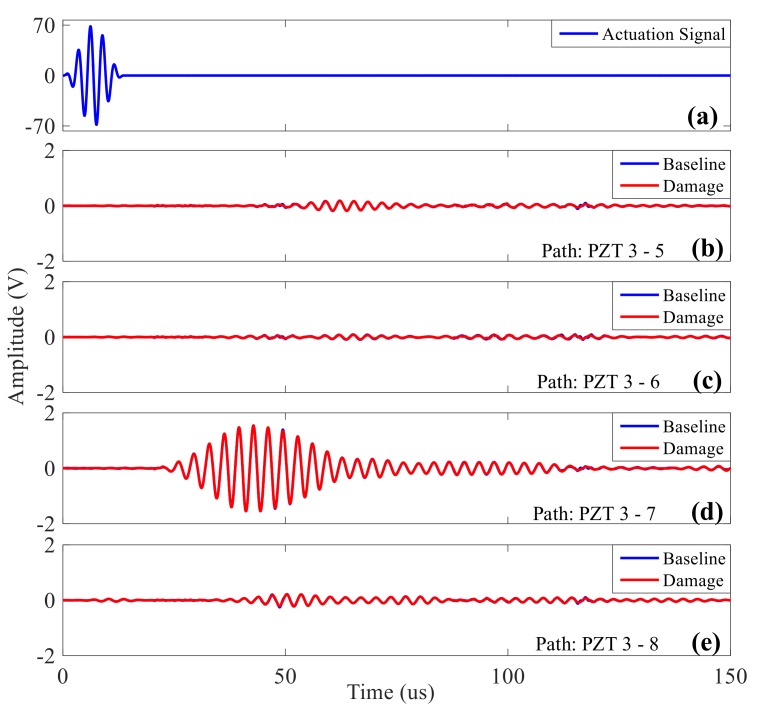
Comparison diagram of reference signal and damage signal in different paths: (**a**) 300 kHz actuation signal; (**b**) PZT three-actuation-five-acquisition; (**c**) PZT three-actuation-six-acquisition; (**d**) PZT three-actuation-seven-acquisition; (**e**) PZT three-actuation-eight-acquisition.

**Figure 16 sensors-20-00898-f016:**
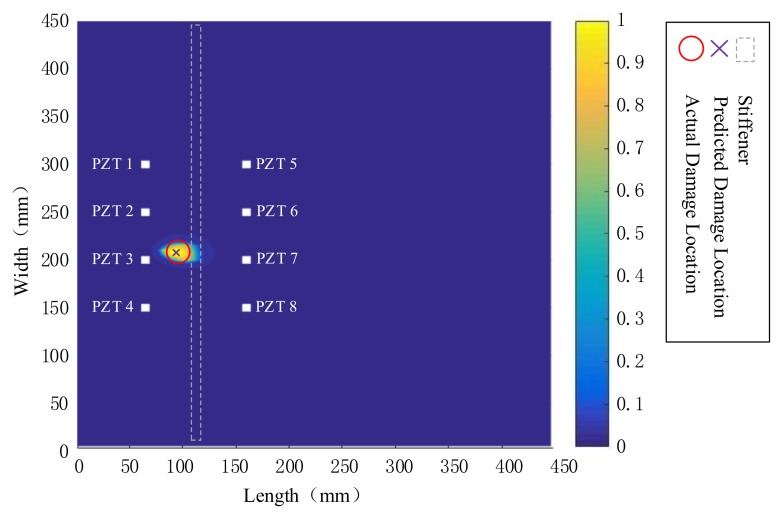
Damage location diagram.

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
