# Peer review of "The Design and Verification of an Active SAMSR Ultrasonic Guided Wave Monitoring System with Ultra-Low Crosstalk"

_sensors, 2020, doi:10.3390/s20030898_

Round 1

Reviewer 1 Report

Ultrasonic guided waves for SHM has been widely concerned since it has a lot of advantage. The paper presents a single-actuation and multiple simultaneous reception circuit system. Most of these works are engineering practice,and the technique in the paper is comparatively common in the phased array ultrasonic instrument. In general, lack of science and innovation in the paper.

The paper work more like a electronic engineering problem, this problem is a common issue and solved well in phased array ultrasonic instrument, the author could get more reference from the traditional phased array ultrasonic instrument development either in industrial intrument or medical ultraonic instrument.

I donot know the journal of "SENSOR" cares more science or general engineering practice, from this point, The paper is doing well for a engineering practice. 

Reviewer 2 Report

Dear Authors,

This paper presents a way of designing multiplexing electronics for a network of multiple ultrasound transducers. The multiplexer has been designed, fabricated and compared with a commercial product for its performance. The experimental results demonstrate the improved attenuation performance. The quality of the paper is good. The reviewer only has the following questions/comments

The reviewer would highly recommend the authors to provide more literature review on the techniques used to isolate pulsing channel from the receiving channel. There are many commercially available products for multi-channel ultrasound transmitting/receiving. Do they all have strong crosstalk as the one from ACELLENT? What is the design difference between the one in this work and the designs used in other commercial products? The authors talked about Lamb wave several times. It is suggested to use "Lamb" rather than "lamb" to be accurate. Also, the term "Lamb signal" is not accurate to the reviewer. Since this is the electrical waveform, it has no relation to Lamb wave which refers to actual acoustic waves. It may be replaced by "tone burst".  On Page 6, the 2nd paragraph, several sentences need to be checked again. It is either a grammar issue or not very clear for its meaning. (1) " An SPDT reed relay has a set of mechanically inter-block..." (2) "...prevents the high voltage actuation signal from acquiring the reception channel and avoids..." (3) "... which reduces the effect of parasitic capacitance is further so that..." On Page 8, the 1st paragraph, the last sentence for diagram (d), the authors stated that the crosstalk can be easily filtered. How could this be filtered? It is of the same frequency of the actual wave signal. 

Author Response

Question 1: The reviewer would highly recommend the authors to provide more literature review on the techniques used to isolate pulsing channel from the receiving channel.

Response: Thanks for the kindly suggestion of the reviewer. As the ultrasonic guided wave system is special in its structure, the proposed two-stage attenuator consisted by reed relays is effective to reduce to crosstalk without reduce the system’s other performance. The research of reducing the electrical crosstalk of ultrasonic guided wave system is rarely, but the mechanical crosstalk is mass. The literature by Fu et al (reference [6] in the paper) has proposed a method to reduce the crosstalk which is compared to our method in this paper. Some literature about crosstalk isolation for high-speed digital board is not direct related to this paper, so they will not be provided.

Question 2: There are many commercially available products for multi-channel ultrasound transmitting/receiving. Do they all have strong crosstalk as the one from ACELLENT? What is the design difference between the one in this work and the designs used in other commercial products?

Response: Thanks for careful comments of the reviewer. We have read some of the literature mentioned in the paper, and tested some commercial ultrasonic guided wave instruments. The strong crosstalk is shown in these products. As the ScanGenie III (made by ACELLENT Ltd., US ) is widely used in the research and industry area due to its advanced function and integration, this paper compares the crosstalk results obtained by our approach with ones obtained by ScanGenie. Meanwhile, some literature has done useful work to reduce the crosstalk in the system. The main difference between the proposed work and the design used in other commercial products is that this paper designs a two-stage attenuation circuit with interlock isolation structure by reed relays instead of using semiconductor switching devices such as high voltage analog switcher of ScanGenie III. The detailed explanation can be found in the paper. (Page 4, the 3nd paragraph)

Question 3: The authors talked about Lamb wave several times. It is suggested to use "Lamb" rather than "lamb" to be accurate. Also, the term "Lamb signal" is not accurate to the reviewer. Since this is the electrical waveform, it has no relation to Lamb wave which refers to actual acoustic waves. It may be replaced by "tone burst".

Response: Thanks for careful comments of the reviewer. We have made corrections in the paper.

Question 4: On Page 6, the 2nd paragraph, several sentences need to be checked again. It is either a grammar issue or not very clear for its meaning. (1) " An SPDT reed relay has a set of mechanically inter-block..." (2) "...prevents the high voltage actuation signal from acquiring the reception channel and avoids..." (3) "... which reduces the effect of parasitic capacitance is further so that..."

Response: Thanks for careful comments of the reviewer. We have made corrections in the paper.

Question 5: On Page 8, the 1st paragraph, the last sentence for diagram (d), the authors stated that the crosstalk can be easily filtered. How could this be filtered? It is of the same frequency of the actual wave signal.

Response: Thanks for careful comments of the reviewer. As the crosstalk is very low, some conventional filter algorithms such as Savitzky-Golay algorithm can be used to smooth the signal which can make the shape and width of the signal remain the same.

Author Response

Question 1: Introduction, Page-1: “As an efficient and flexible online/offline damage inspection and diagnosis method, -------------- connected structures”. Only one reference (application in rails) is cited to show the application of guided waves. Authors are suggested to cite more recent references demonstrating guided wave applications.

Response: Thanks for the kindly suggestion of the reviewer. We have added the new recent references in the introduction.

Question 2: The term “Hot spot regions” is used many times. However, it is not explained properly in the manuscript.

Response: Thanks for the kindly suggestion of the reviewer. The hot spot regions refer to the structural parts or regions with high stress, such as bolted and adhesive joints. We have explained in Page 1, section 1, the 1st paragraph.

Question 3: Page 10: There is no information about the composite material. The type and properties of composite material must be included.

Response: Thanks for the kindly suggestion of the reviewer. The composite material plate is made of 15 plies of T300 woven prepreg with a process of vacuum bag molding. We have added these information in Page 11, the 1st line.

Question 4: Page 11: What type of PZTs you used?

Response: Thanks for the careful comments of the reviewer. The type of PZTs we used is PZT-5A and we have added these information in Page 11, the 1st paragraph. The material parameters of PZT-5A are listed in the following table.

Material parameters of the PZT-5A.

Type

Density (kg/m3 )

Curie Temp (℃)

Diameter (mm)

Thickness (mm)

Dielectric Constant

d31

d33

PZT-5A

7750

360

8

0.44

1700

-200

530

Question 5: Each system has limitations. There are many ultrasonic systems developed. However the defect estimation and detectability also depend on the type and complexity of structure under investigation and cross-talk in some cases cannot be removed. Postprocessing/signal processing is required in most of the cases. The authors must include the limitation of the developed system. What is the scope of improvement?

Response: Thanks for the careful comments of the reviewer. The limitation of the developed system includes two parts. One is that the system integration is not high enough since the reed relays (DIP05-1C90) used for the 2two-stage attenuator seems to be a little big. The other is that the shielded wire to connect the PZTs is required to reduce the crosstalk well. As mentioned in Section 5, the higher integration can be achieved by optimizing the selection of chip devices further.

Question 6: Page 13: What is the size of the damage? What is the accuracy in the location and size of damage?

Response: Thanks for the careful comments of the reviewer. The size of the simulated damage is about Φ25 mm as shown in figure 16, which is added into the page 13, the 1st line of revised manuscript . The accuracy in the location of damage can be evaluated by absolute error. The absolute error is defined as the distance between the actual location and the imaged location. While the size of damage can be estimated in a way of calibration. The current study just considers the accuracy of location of damage.

Reviewer 4 Report

The paper describes the design and application of phased-array ultrasound with low crosstalk between actuation and sensing channels by using reed switching. My comments are:

The authors should reference earlier work describing reed switching in phased-array ultrasound, for example Waag and Fedewa (IEEE Trans. Ultrason., Ferroelect., Freq. Contr., 53, pp. 1707-1718) 2006. The novel content of this paper should be emphasised in comparison with prior research. Page 1, section 1, line 4. The frequency response of the PZTs should be chosen to provide a balance between a short pulse time, allowing accurate time of flight measurement, and wave dispersion in the material at different frequencies. Page 2, line 15. Check wording ‘… a purity mode Lamb waves …’ Page 2, line 15. The blind area for damage detection due to crosstalk should be explained. Page 3, section 2.1, lines 5-6. ‘5 waves peaks’ is better expressed as ‘a wavepacket containing 5 peaks’. Is a windowing function used? Page 3, last 2 lines and first 3 lines of page 4. This text appears to be included by mistake. Page 5, figure 3. The ‘Damage signal’ waveforms are not as expected. In PZT2 and PZT5 signals the second blue wavepacket is missing. Also the PZT2 and PZT4 signals should have a short delay between the blue and red wavepackets, and the PZT3 and PZT5 signals should have a longer delay between the blue and red wavepackets. Page 5, section 3.1, lines 5 and 6. The signal generator output is an electrical waveform, not a Lamb wave. The Lamb wave is only present in the plate. Page 8, figure 6. The circuit design achieves a good reduction of crosstalk (from 3% to 0.04%) compared with the commercial system. Page 10, line 2. The model number and manufacturer should be given for the CPU card, signal generator card and oscilloscope card. Page 10, first line after figure 10. What is the height of the T-stiffener? Page 13, line 1. Why was the elliptic weighting algorithm selected? Page 13, figure 16. The colorbar does not have any units.

Author Response

Question 1: The authors should reference earlier work describing reed switching in phased-array ultrasound, for example Waag and Fedewa (IEEE Trans. Ultrason., Ferroelect., Freq. Contr., 53, pp. 1707-1718) 2006. The novel content of this paper should be emphasised in comparison with prior research.

Response: Thanks for careful comments of the reviewer. This paper is about ultrasonic guided wave monitoring system, which is a little similar with the phased array ultrasonic system. While compared the ultrasonic guided monitoring system to the phased array ultrasonic system, there are two important differences. First, the phased array ultrasonic system use the time delay compensation to enhance the echo signal, but the ultrasonic guided monitoring system use the difference between baseline and damage signals to find damage areas. All of the elements start to collect echo signals after the end of actuation signal in the phased array system, so the strong crosstalk generated by actuation signal would not affect the system detection accuracy, which can be ignored directly. However, in our system, the echo or scatter signal is collected at the same time of actuation signal starting, so the strong crosstalk may cause blind detect areas. Second, in the phased array system, all the elements are set an independent A/D converter channel to realize all receive channels simultaneous acquisition. Therefore, the whole hardware system is expensive due to many converters. But in the ultrasonic guided monitoring system all the receive channels can shared one or several A/D converter channels by applying matrix switcher, which is designed in the paper additionally. The literature referred used two banks of reed relays to consist the multiplexer. In the paper, the reed relays is mainly used to consist of the two-stage attenuator which can realize the ultra-low crosstalk performance. The reed relays also play a role of the multiplexer in actuation channels, but it is better to replace by high voltage analog multiplexer just like the commercial instruments if not consider the attenuator function. As the ultrasonic guided wave system is special in its structure, the proposed two-stage attenuator based on reed relays is effective to reduce to crosstalk without reducing the system’s other performance. The research of reducing the electrical crosstalk of ultrasonic guided wave system is rare, but the mechanical crosstalk is mass. The literature by Fu et al (reference [6] in the paper) has proposed a method to reduce the crosstalk which is compared to our method in this paper.

Some introduction of the difference between the two kinds of the system is added in Page 2, the 2nd paragraph.

Question 2: Page 1, section 1, line 4. The frequency response of the PZTs should be chosen to provide a balance between a short pulse time, allowing accurate time of flight measurement, and wave dispersion in the material at different frequencies.

Response: Thanks for careful comments of the reviewer. The damage diagnostic algorithm used in the paper use the difference between baseline and damage signals to find damage areas, which is different from phased array ultrasonic system. The delay time compensation is not needed, as well as the accurate time of flight.

Question 3: Page 2, line 15. Check wording ‘… a purity mode Lamb waves …’

Response: Thanks for the careful comments of the reviewer. We have made corrections in Page 2, the 1st paragraph. A purity mode Lamb wave has been revised into a pure mode Lamb wave.

Question 4: Page 2, line 15. The blind area for damage detection due to crosstalk should be explained.

Response: Thanks for the kindly suggestion of the reviewer. Due to the existence of crosstalk, the sensor's first arrival signal will be superimposed with the crosstalk signal. If the distance between actuator and sensor is very short in the case of hot spot monitoring, there is a blind area for damage detection. We have explained in Page 2, the 1st patagraph..

Question 5: Page 3, section 2.1, lines 5-6. ‘5 waves peaks’ is better expressed as ‘a wavepacket containing 5 peaks’. Is a windowing function used?

Response: Thanks for the kindly suggestion of the reviewer. The 5-cycle Hanning windowed toneburst are used to excitation signals. We have made corrections and added the relative information in Page 3, section 2.1.

a 5-cycle Hanning windowed toneburst

Question 6: Page 3, last 2 lines and first 3 lines of page 4. This text appears to be included by mistake.

Response: Thanks for the careful comments of the reviewer. It is a mistake and we have deleted the text.

Question 7: Page 5, figure 3. The ‘Damage signal’ waveforms are not as expected. In PZT2 and PZT5 signals the second blue wavepacket is missing. Also the PZT2 and PZT4 signals should have a short delay between the blue and red wavepackets, and the PZT3 and PZT5 signals should have a longer delay between the blue and red wavepackets.

Response: Thanks for the careful comments of the reviewer. Our original purpose is going to use the figure 3 to show damage signal is different, so they are not real damage signals. Strictly speaking, they are wrong, and we have made corrections in figure 3.

Question 8: Page 5, section 3.1, lines 5 and 6. The signal generator output is an electrical waveform, not a Lamb wave. The Lamb wave is only present in the plate.

Response: Thanks for the careful comments of the reviewer. The Lamb wave is only present in the plate just like you said. We have used “tone burst” instead of “lamb wave”.

Question 9: Page 10, line 2. The model number and manufacturer should be given for the CPU card, signal generator card and oscilloscope card.

Response: Thanks for the kindly suggestion of the reviewer. The CPU card is NI PXIe-8820, the signal generator card is NI PXIe-5442 and the oscilloscope card is NI PXIe-5105. We have added these information in Page 10, section 4, the 1st paragraph.

Question 10: Page 10, first line after figure 10. What is the height of the T-stiffener?

Response: Thanks for the careful comments of the reviewer. The height of the T-stiffener is 40mm. We have added in Page 10, section 4 , the 2nd parapragh.

Question 11: Page 13, line 1. Why was the elliptic weighting algorithm selected?

Response: Thanks for the careful comments of the reviewer. This paper would not focus to the algorithm of analyze the damage location. The reason why the elliptic weighting algorithm was selected is that this algorithm does not require the calculation the time of flight of Lamb wave that is very difficult for Anisotropic composite plate. Meanwhile, our group has done some research on this algorithm.

Question 12: Page 13, figure 16. The colorbar does not have any units.

Response: Thanks for the careful comments of the reviewer. The colorbar represents the probability of damage at that point, so it has no unit.

Round 2

Reviewer 1 Report

Crosstalk is a very important problem for multi channels ultrasonic inspection instruments. In fact, for an ultrasonic guided wave monitoring system which composed of transducer network, pulse-echo and pitch-catch mode are both needed. Crosstalk which exists in pulse-echo mode is more severe and more difficult to solve than in the pitch-catch mode. At the beginning of the paper, it should be clear that this article only aims at the crosstalk of pitch-catch mode, which is not easy to be misunderstood. Increasing detection efficiency by increasing the acquisition channels is not creativity, since it is a very common and obvious method, and it will increase the cost. The highlight of this paper is the design of two-stage relay switching circuit. Since the lifetime of the type of reed relay is limited, how about it affects the monitoring period of the system? Figure 12 is the test result for the T-shaped stiffener in Figure.11 or the plate in Figure.3? It does not make sense. In general, this paper focuses on electronic engineering implementation for a multi-channel switch circuit system. It is significant for the practical application of guided wave monitoring. I think the paper can be accepted by some revision.

Author Response

Question 1: At the beginning of the paper, it should be clear that this article only aims at the crosstalk of pitch-catch mode, which is not easy to be misunderstood. Increasing detection efficiency by increasing the acquisition channels is not creativity, since it is a very common and obvious method, and it will increase the cost. The highlight of this paper is the design of two-stage relay switching circuit.

Response: Thanks for the kindly suggestion of the reviewer. We have made corrections to emphasize the work to solve the problem of crosstalk of pitch-catch mode in the Abstract Section and Page 3, the 2nd paragraph.

Question 2: Since the lifetime of the type of reed relay is limited, how about it affects the monitoring period of the system?

Response: Thanks for the kindly suggestion of the reviewer. As analyzed in Page 7, Section 3.2, the 1st paragraph, the maximum switching time of the high speed reed relays (DIP05-1c90) used in first stage attenuator is less than 1.5ms while operating in max current load according to the datasheet, and the reed relays (CRR05) used in second attenuator is less than 0.6ms. The two stage of attenuator’s reed relays are controlled in parallel by the same control signal, so increasing the stage of attenuators will not increase the switch time. In the scanning process of the ultrasonic detection system, the delay of milliseconds between the two excitation signals is required to wait for the echo to disappear, so the reed relays switching speed would almost not affect the damage scanning cycle. In fact .while the reed relays is switching, there is barely not current and voltage in the contact, because the actuation signal is started after the reed relays switching ready. Moreover, the reed relay contacts are protected by inert gas, preventing it from oxidizing and giving it long contact life. Therefore, the reed relays will switch much faster than their maximum switching time with long lifetime. This paper is only propose a realizing method, and there are much higher speed reed relays can be selected.

Question 3: Figure 12 is the test result for the T-shaped stiffener in Figure.11 or the plate in Figure.3? It does not make sense.

Response: Thanks for the careful comment of the reviewer. The Figure 12 test result is not related to Figure 11 or Figure 3. It is the result when the actuation and the reception sensors are close, which is not shown in this paper as it can be obtained by a conventional experiment. This Figure want to show the echo signals can be fully displayed by the developed system in this paper as the crosstalk is ultra-low.

Reviewer 3 Report

The authors have addressed all comments/suggestions in the present form of the manuscript. Therefore, I would recommend to accept it for publication in the Journal of "Sensors".

Author Response

Thanks again for the kindly comments/suggestions.